# A Novel Role of BIRC3 in Stemness Reprogramming of Glioblastoma

**DOI:** 10.3390/ijms23010297

**Published:** 2021-12-28

**Authors:** Qiong Wu, Anders E. Berglund, Robert J. MacAulay, Arnold B. Etame

**Affiliations:** 1Department of Neuro-Oncology, H. Lee Moffitt Cancer Center and Research Institute, Tampa, FL 33612, USA; qiong.wu@moffitt.org; 2Department of Biostatistics and Bioinformatics, H. Lee Moffitt Cancer Center and Research Institute, Tampa, FL 33612, USA; anders.berglund@moffitt.org; 3Departments of Anatomic Pathology, H. Lee Moffitt Cancer Center and Research Institute, Tampa, FL 33612, USA; robert.macaulay@moffitt.org

**Keywords:** brain tumor, GBM, cancer stem cell, BIRC3, BMP4, stemness

## Abstract

Stemness reprogramming remains a largely unaddressed principal cause of lethality in glioblastoma (GBM). It is therefore of utmost importance to identify and target mechanisms that are essential for GBM stemness and self-renewal. Previously, we implicated BIRC3 as an essential mediator of therapeutic resistance and survival adaptation in GBM. In this study, we present novel evidence that BIRC3 has an essential noncanonical role in GBM self-renewal and stemness reprogramming. We demonstrate that BIRC3 drives stemness reprogramming of human GBM cell lines, mouse GBM cell lines and patient-derived GBM stem cells (GSCs) through regulation of BMP4 signaling axis. Specifically, BIRC3 induces stemness reprogramming in GBM through downstream inactivation of BMP4 signaling. RNA-Seq interrogation of the stemness reprogramming hypoxic (pseudopalisading necrosis and perinecrosis) niche in GBM patient tissues further validated the high BIRC3/low BMP4 expression correlation. BIRC3 knockout upregulated BMP4 expression and prevented stemness reprogramming of GBM models. Furthermore, siRNA silencing of BMP4 restored stemness reprogramming of BIRC3 knockout in GBM models. In vivo silencing of BIRC3 suppressed tumor initiation and progression in GBM orthotopic intracranial xenografts. The stemness reprograming of both GSCs and non-GSCs populations highlights the impact of BIRC3 on intra-tumoral cellular heterogeneity GBM. Our study has identified a novel function of BIRC3 that can be targeted to reverse stemness programming of GBM.

## 1. Introduction

Glioblastoma (GBM) is a highly resistant and lethal brain cancer with limited treatment options. The current multimodal therapy of maximal-safe surgical resection, radiation therapy (RT) and concurrent temozolomide (TMZ) leads to a median survival of only 14 months [1]. A classic hallmark of GBM is the rapid acquisition of therapeutic resistance leading to lethality. There is therefore a significant unmet need for effective anti-GBM therapies that prevent early acquisition of resistance.

Stemness adaptation is a leading hypothesis for therapeutic failures in GBM [2]. GBM cells with stem-like phenotype known as GBM stem-like cells or GBM stem cells (GSCs) drive resistance to RT and TMZ treatment [3]. Evidence from GBM mouse models implicates GSCs in the repopulation of tumors following TMZ and RT treatment [4]. Plasticity towards tumor repopulation is the basis for recurrence, cellular intra-tumoral heterogeneity and disease progression. GSCs have tumor-initiating capabilities and can induce tumors in vivo that recapitulate the molecular features of the parental GBM tumor [4,5,6]. Stemness phenotype in GBM is characterized by expressions of CD133 and the neural stem cell marker Nestin [7,8,9,10,11]. From a transcriptomics perspective, GSCs exploit normal neural stem cell developmental transcription mechanisms towards tumorigenesis adaptation [12,13,14]. Given the central role of stemness reprogramming in GBM resistance, it is therefore of utmost importance to identify and target essential stemness reprogramming mechanisms in GBM. Strategies that prevent or reverse stemness reprogramming would undoubtedly have a significant impact in GBM.

Previously, we implicated the anti-apoptotic protein, BIRC3 as an essential mediator of therapeutic resistance and survival adaptation in GBM [15,16]. BIRC3 is an inhibitor of apoptosis protein with established canonical anti-apoptosis function through inhibition of caspase activation [17,18]. Using GBM cell lines and GBM patient tissue samples, we have established that BIRC3 contributed toward TMZ and RT resistance in GBM through PI3K and STAT3 signaling activation [15]. We also showed that BIRC3 expression increased during GBM treatment, GBM recurrence and adversely impacted upon GBM patient survival [15]. In support of our findings, another group independently implicated BIRC3 as a facilitator of malignant progression in GBM [19]. In a subsequent study, we reported that BIRC3 was an important contributor to GBM hypoxia adaptation and mesenchymal phenotype [16]. Although the preponderance of evidence thus far supports a role for BIRC3 in GBM resistance adaptation, it remains unclear if BIRC3 has any role in GBM stemness adaptation.

Based on our previous discoveries of several novel noncanonical functions of BIRC3 in GBM survival adaptation, we hypothesized that BIRC3 was also critical for stemness reprogramming. In support of this hypothesis, we present novel evidence that BIRC3 regulates stemness reprogramming, tumor initiation and tumor progression in GBM. Importantly, we demonstrate that BIRC3 drives stemness reprogramming of human GBM cell lines, mouse GBM cell line and patient-derived GSCs through regulation of BMP4 signaling axis. RNA-Seq interrogation of the stemness reprogramming hypoxic (pseudopalisading necrosis and perinecrosis) niche in GBM patient tissues further validated the high BIRC3/low BMP4 expression correlation. Our findings represent the first implication of an anti-apoptotic protein in GBM cell-fate stemness reprogramming. We have therefore identified a novel noncanonical function of BIRC3 that can be targeted to reverse stemness programming of GBM.

## 2. Results

### 2.1. BIRC3 Expression Correlates with Stem Cell Markers Expression and Self-Renewal in Both Human and Mouse GBM Cells

We previously validated BIRC3 as a novel anti-GBM target for therapeutic resistance [15]. We were therefore interested in determining if BIRC3 played a role in GBM cell fate stemness reprogramming. We therefore established BIRC3 overexpressing and BIRC3 knockout lines in U251 and U87 human GBM cell lines. BIRC3 protein expression was validated by western blot (Figure 1A). In order to evaluate the impact of BIRC3 on stemness reprogramming, we evaluated the impact of BIRC3 gain-of-function and loss-of-function on the ability of GBM cells to form neurospheres. GSCs grow as neurospheres and therefore neurosphere formation serves as a surrogate for stemness [20]. Equal cell numbers were seeded into wells and allowed to grow into colonies under stem cell culture conditions (U251: 5000 cells/well; U87: 2000 cells/well). BIRC3 overexpression significantly enhanced neurosphere formation in both U251 and U87 GBM cells (Figure 1B,C, *p* < 0.05). Moreover, we observed a significant reduction in neurosphere formation in U87 BIRC3 knockout GBM cells compare to control cells (Figure 1C, *p* < 0.05). In U251 GBM cells, BIRC3 knockout significantly impacted neurosphere formation but to a lesser extent compared to U87 GBM cells (Figure 1B, *p* < 0.05). To further determine the effect of BIRC3 on maintenance of GBM stemness, we examined two different stem cell gene markers *CD133* and *ABCG2* expression by real-time PCR [21]. CD133 is a glycoprotein that is the most employed marker for isolation of cancer stem cell population from different tumors, especially various gliomas [7,8,10,22]. ABCG2 (ATP-binding cassette super-family G member 2) is a membrane-associated protein also a known cancer stem cell marker in gliomas [23,24,25]. Interestingly, BIRC3 overexpression significantly induced CD133 and ABCG2 expressions in both U251 and U87 cell lines (Figure 1D, *p* < 0.05). BIRC3 knockout cells was associated with a significant reduction in both CD133 and ABCG2 expressions compared to control wild type cells (Figure 1D, *p* < 0.05). Moreover, we also examined another stemness marker ALDH1A3. ALDH1A3 (Aldehyde dehydrogenase 1 family member A3) an isozyme metabolizes aldehydes to their respective carboxylic acid and is higher expressed in the cancer stem cell niche of GBM and other cancers [26,27]. In both U251 and U87 GBM cells, BIRC3 overexpression significantly increased ALDH1A3 expression compared to wild type control cells, while BIRC3 knockout only reduced ALDH1A3 expression in U87 cells (Appendix A, *p* < 0.05). To further characterize the role of BIRC3 in GBM stemness reprogramming, we evaluated Nestin expression via confocal immunocytochemistry analysis. Nestin expression is a prerequisite for the maintenance of stemness [10,11]. There was a direct and strong association between BIRC3 and Nestin expressions in both U251 and U87 GBM cells whereby BIRC3 overexpression induced higher Nestin expressions compared to BIRC3 knockout cells (Figure 1E). In order to validate our findings in a mouse GBM cell model, we repeated these experiments using CT-2A cell line, which is a murine glioma cell line [28]. BIRC3 expression enhanced neurosphere formation in CT-2A cells as we had observed in the human glioma cell lines. Mouse BIRC3 protein expression and knockout efficiency were validated by western blot (Figure 1F). BIRC3 overexpression significantly increased neurosphere formation capacity (Figure 1G, *p* < 0.05) while BIRC3 knockout significantly reduced neurosphere formation capacity (Figure 1G, *p* < 0.05). Importantly, CD133 and ABCG2 were significantly upregulated in BIRC3 overexpressed cells and downregulated in BIRC3 knockout cells (Figure 1H, *p* < 0.05). Collectively, these findings suggest that BIRC3 expression is critical for GBM cell self-renewal and stemness maintenance.

### 2.2. Human GBM Stem Cell Self-Renewal Is Regulated by BIRC3 Expression

In an effort to ascertain if BIRC3 had a similar impact on human GBM stem cells (GSCs), we established BIRC3 overexpression and BIRC3 knockout lines in three patient-derived GSCs (Figure 2A). First, we examined the effect of BIRC3 expression on GSC stemness maintenance by evaluating CD133 and ABCG2 expression levels in GSCs. Real-time PCR results revealed that BIRC3 overexpression significantly induced higher expressions of CD133 in GSC-2 (Figure 2B, *p* < 0.05); and higher expressions of ABCG2 in both GSC-1 and GSC-2 to maintain self-renewal and stemness (Figure 2B, *p* < 0.05). Interestingly, CD133 and ABCG2 expressions were significantly downregulated in all BIRC3 knockout GSCs including GSC-1, GSC-2 and GSC-3 (Figure 2B, *p* < 0.05). Moreover, ALDH1A3 expression was significantly downregulated in all BIRC3 knockout GSCs compared to wild type control, while its expression was induced only in BIRC3 overexpressed GSC-1 and GSC-3 (Appendix A). Next, in order to further understand and validate the stemness phenotype induced by BIRC3, we cultured all three GSCs under differentiating media conditions. BIRC3 overexpression facilitated formation of neurospheres and self-renewal in differentiated GSCs, whereas BIRC3 knockout prevented neurosphere formation and self-renewal (Figure 2C, *p* < 0.05) in differentiated GSCs. A similar trend was noted with real-time PCR analysis of CD133 and ABCG2 expressions. A significant fraction of BIRC3 overexpression GSCs demonstrated enhanced stemness marker expression; enhanced self-renewal capabilities; and enhanced stemness maintenance (Figure 2D, *p* < 0.05). BIRC3 knockout significantly inhibited stemness marker expression; self-renewal capabilities; and stemness maintenance compared to wild type cells in GSC-2 and GSC-3. Moreover, BIRC3 overexpression significantly increased ABCG2 expression, while BIRC3 knockout significantly suppressed ABCG2 expression (Figure 2D, *p* < 0.05). Furthermore, confocal immunocytochemistry analysis revealed that BIRC3 expression was sufficient in strongly inducing Nestin expression in all 3 GSCs (Figure 2E). Therefore, BIRC3 serves as a critical regulator in GSC self-renewal and stemness maintenance, even in differentiated GSCs.

### 2.3. BIRC3 Regulates BMP4 Signaling Inhibition in GBM

BMP4 is strongly associated with GBM stem cell differentiation [29,30,31], and has been reported as a potential anti-GBM target [32]. Since BIRC3 expression is associated with GBM cell self-renewal and stemness maintenance, we wanted to determine if there was any correlation between BIRC3 expression and BMP4 signaling activation. We sought to examine the relationship between *BIRC3* and *BMP4* using patient GBM tissue data. We initially estimated mRNA expression correlation between *BIRC3* and *BMP4* using TCGA (The Cancer Genome Atlas) GBM PanCan dataset. This dataset contains whole tumor data and therefore does not account for regional heterogeneity in GBM. *BIRC3* and *BMP4* had a low correlation in GBM PanCan dataset (Appendix A). However, considering the GBM intra-tumoral heterogeneity and BIRC3 regional expression [16], we then analyzed BIRC3 and BMP4 expression using the IVY Glioblastoma Atlas dataset that has RNA-Seq datasets from MRI-distinct GBM regions. We examined hypoxic (pseudopalisading necrosis and perinecrosis) and vascular (hyperplastic blood vessels and vascular proliferative) regions of GBM. The hypoxic region has been established as a critical niche for stemness reprogramming in GBM [33,34,35,36]. There was a negative correlation between *BIRC3* and *BMP4* expressions in both the vascular and hypoxic niches in GBM (Pearson r = −0.475; spearman r = −0449; Figure 3A). The stemness reprogramming hypoxic niche demonstrated high BIRC3/low BMP4 expression profile. Conversely, the vascular niche demonstrated low BIRC3/high BMP4 expression profile. To further confirm if the impact of BIRC3 on GBM cell stemness and self-renewal is correlated with BMP4 expression, we first performed real-time PCR analysis to evaluate BMP4 expression in both BIRC3 overexpressing and knockout GBM cell lines. We found that BIRC3 overexpression significantly inhibited BMP4 expression in GBM cell lines compare to wild type control. Interestingly, we found that knockout of BIRC3 significantly activated BMP4 expression (Figure 3B, *p* < 0.05). Moreover, similar observations were made in undifferentiated GSCs, differentiated GSCs and CT-2A mouse GBM cells (Figure 3C–E). BIRC3 significantly suppressed BMP4 expression in GSC-2 and GSC-3; and, furthermore, depletion of BIRC3 induced BMP4 expression in GSC-1 and GSC-2 (Figure 3C, *p* < 0.05). In differentiated GSCs, high levels of BIRC3 significantly inhibited BMP4 expression (Figure 3D, *p* < 0.05). However, depletion of BIRC3 increased BMP4 expression only in GSC-2 (Figure 3D, *p* < 0.05). We observed a similar gene expression pattern in CT-2A cells as well as human GBM cell lines (Figure 3E, *p* < 0.05). Furthermore, western blot results indicated that in BIRC3 knockout U251 and U87 cells, BMP4 signaling was strongly activated through SMAD1/5 phosphorylation and BIRC3 overexpression suppressed this activation effectively (Figure 3F). These results suggested that BIRC3 could directly suppress BMP4 signaling activation in GBM cell lines and stem cells.

### 2.4. BIRC3 Mediated Stemness Reprogramming in GBM Cells Is Dependent on BMP4 Suppression

Since BIRC3 knockout inhibits GBM cell self-renewal, we were interested in determining if this process was directly driven by BMP4 signaling activation. When we silenced BMP4 in BIRC3 knockout U251 and U87 cells using selective siRNAs, we observed a significant blockade and reversal of low-BIRC3 induced up-regulation of BMP4 expression (Figure 4A, *p* < 0.05). Moreover, we were also interested in determining if silencing BMP4 in BIRC3 knockout cell could restore GBM cell stemness reprograming. We therefore first examined the relative expressions of stemness markers CD133 and ABCG2 following BMP4 silencing. In BIRC3 knockout U251 cells, siRNA silencing of BMP4 in BIRC3 knockout cells significantly increased CD133 and ABCG2 expressions more than 2-fold (Figure 4B, *p* < 0.05). A similar trend in CD133 and ABCG2 expressions was noted in U87 cells (Figure 4C, *p* < 0.05). Interestingly, silencing of BMP4 in BIRC3 knockout U251 and U87 cells reduced SMAD1/5 phosphorylation, which had initially been induced by BIRC3 knockout, and further suppressed BMP4 signaling activation (Figure 4D). Next, we performed tumor sphere assay to determine if BMP4 signaling suppression directly impacted upon GBM cell self-renewal phenotype and neurosphere formation. The tumor sphere formation assay results revealed that silencing BMP4 in BIRC3 knockout U251 cells significantly induced tumor sphere formation and rescued the loss of stemness which was previously induced by BIRC3 knockout (Figure 4E, *p* < 0.05). These results suggested that low expression of BIRC3 suppresses GBM cell self-renewal through BMP4 signaling activation and that silencing of BMP4 could significantly restore stemness. Hence, BIRC3 drives stemness reprogramming in GBM through suppression of BMP4 signaling activation.

### 2.5. BIRC3 Influences Tumor Initiation and Progression in GBM Orthotopic Xenograft Model

We wanted to determine if our in vitro data on BIRC3 stemness reprogramming phenotype had any in vivo relevance. We established orthotopic intracranial mouse xenografts consisting of including wild-type, BIRC3 overexpression and BIRC3 knockout U251 GBM cells. Tumor cells were stereotactically implanted into the brains of mice. Intracranial xenografts were monitored with MRI for tumor formation and progression. The MRI results indicated that BIRC3 expression significantly facilitated GBM tumor initiation and progression, whereas BIRC3 knockout significantly inhibited tumor initiation and progression (Figure 5A,B). Kaplan-Meier survival curve was recorded at desired time points (Figure 5C, *n* = 5). BIRC3 overexpression accelerated tumor progression and significantly decreased survival (Figure 5C, *p* < 0.023), while BIRC3 knockout significantly increased survival (Figure 5C, *p* = 0.00008). We further examined the xenograft tissues by H&E staining and immunohistochemistry for BIRC3. We confirmed a marked high expression of BIRC3 in BIRC3 overexpressing U251 GBM xenograft compared to wild-type control U251 GBM xenograft (Figure 5D). To validate BIRC3 modulation of stemness genes in vitro, we further measured relative expression of CD133, ABCG2 and BMP4 using mRNA samples extracted from xenograft tumor tissues. BIRC3 overexpression significantly induced both CD133 and ABCG2 expressions, and reduced BMP4 expression (Figure 5E, *p* < 0.05), while BIRC3 knockout resulted in the opposite effect (Figure 5E, *p* < 0.05). Taken together, these data suggest that BIRC3 could impact on GBM tumor initiation, stemness and progression.

## 3. Discussion

GBM is highly lethal cancer largely due to persistence and propagation of GSCs with enhanced stemness phenotype despite TMZ and RT. Hence identifying mechanisms of GBM stemness is very important in advancing our understanding and targeting of GBM resistance. In this study, we report the novel discovery that BIRC3 expression promotes GBM stemness and tumorigenicity of GSCs through inactivation of BMP4 signaling pathway. Using a combination of GBM cell lines, patient-derived GSCs and GBM patient tissue regional RNA-Seq data, we established the association between BIRC3 expression and GBM stemness maintenance. Specifically, we demonstrate that BIRC3 induced stemness and self-renewal through downstream inactivation of BMP4 signaling. Furthermore, the loss of stemness associated with BIRC3 knockout can be reversed or rescued through siRNA silencing of BMP4 signaling. Lastly, we demonstrate that depletion of BIRC3 significantly suppressed tumor initiation and progression in GBM intracranial xenografts. Our discovery reveals a novel function of BIRC3 that has never been described and that appears to be independent of the canonical anti-apoptotic functions of BIRC3 in GBM. Our findings therefore have several significant implications.

BMP4 signaling has been implicated in GSC differentiation and inhibition of GSC self-renewal and tumorigenicity [29,30,31]. BMP4 suppresses CD133 expression and CD133-positive GSC populations [30,31]. Furthermore, it has been reported BMP4 could inhibit GSC self-renewal and tumorigenicity through SMAD1/5 phosphorylation [29]. Hence BMP4 is an important driver of GSC differentiation and loss of stemness. Interestingly, in both our in vitro and in vivo studies, it appears that BIRC3 is a critical negative regulator of BMP4 signaling activation in GBM. In this role BIRC3 can therefore directly impact upon GBM cell self-renewal and differentiation. Hence BIRC3 emerges as a robust GBM stemness regulator. Our work for the first time also shows that high BIRC3 expression could significantly induce GBM cell self-renewal and stemness maintenance. We further present new evidence that depletion of BIRC3 significantly enhances activation of BMP4-SMAD1/5 signaling in GBM.

The identification of elevated BIRC3 expression as a GBM stemness maker is novel. Interestingly, our analysis indicates that BIRC3 is an independent biomarker for stemness not only in human/mouse GBM cell lines but also patient-derived GSCs. Our data indicated that BIRC3 could significantly increase neurosphere formation ability in both human and mouse GBM cell lines and patient-derived GSCs. Even upon differentiation of patient-derived GSCs, subsequent upregulation of BIRC3 restored GSCs self-renewal and stem-like phenotype. Hence BIRC3 contributed to stemness even in differentiated GBM cells. A major implication of this finding is that through propagation of stemness in both GSCs and non-GSCs populations, BIRC3 appears to be a major driver of intra-tumoral cellular heterogeneity in GBM. We previously demonstrated that BIRC3 was upregulated in GBM recurrence, TMZ-resistance, RT treatment and GBM hypoxia [15,16]. Our current study would imply that upregulation of BIRC3 in the above context is a central mechanism for treatment and microenvironment induced stemness reprogramming.

Moreover, our results suggest that BIRC3 could maintain GBM cell self-renewal and stemness through inhibiting BMP4 expression and further inactivating downstream SMAD1/5 phosphorylation. Suppression of BMP4 signaling will result in cell differentiation inhibition and expression of GBM stem cell marker CD133 and ABCG2. Hence, within the context of BIRC3/BMP4 axis, BMP4 antagonizes BIRC3-induced stemness in GBM. This is further supported by the observation that depletion of BMP4 in BIRC3 knockout cells could significantly restore stemness. Interestingly, high *BIRC3* expression correlated with low *BMP4* expression within the hypoxic niche. Given the critical role of the hypoxic niche in GBM stemness reprogramming [33,34,35,36], our findings provide further support for the hypothesis that BIRC3/BMP4 axis regulates stemness reprograming in GBM. Further studies are necessary to fully understand the molecular underpinnings of BMP4 pathway signaling with respect to BIRC3.

Lastly, we evaluated the impact of BIRC3 on tumor initiation. We demonstrated in orthotopic intracranial xenografts that high BIRC3 expression could significantly promote tumor initiation and propagation and moreover BIRC3 depletion could enhance survival though suppressing of tumor initiation and growth. We believe this is in line with the impact of BIRC3 on GBM stemness. The strategy of preventing tumor initiation through depletion of BIRC3 is of clinical importance and addresses a major reason for treatment failures in GBM.

In summary, our studies have shed some lights with respect to the regulation of GBM stem cell self-renewal and stemness maintenance. In particular, a novel translational function of BIRC3 in GBM and GSCs has been uncovered. Our data supports targeting BIRC3/BMP4 axis as a relevant therapeutic approach in addressing GBM stemness reprogramming. Further mechanistic elucidation of BIRC3/BMP4 signaling will undoubtedly provide new therapeutic avenues for GBM patients.

## 4. Methods and Materials

### 4.1. Cell Culture and Reagents

U251 and U87 human glioblastoma cell lines (ATCC) were cultured in DMEM (Life Technologies) supplemented with 10% fetal bovine serum (Sigma-Aldrich, St. Louis, MO, USA), 100 units/mL penicillin and -100 ug/mL streptomycin (Life Technologies, NY, USA). The cultures were maintained at 37 °C in a humidified atmosphere containing 5% CO_2_. The patient-derived GSCs used in this study were isolated from GBM patients and were well characterized. The patient-derived GSCs were culture in NS-A medium (90% NeuroCult NS-A Basal Medium Human plus 10% Human NeuroCult NS-A proliferation Supplements, StemCell Technologies). Complete medium was supplied with recombinant human epidermal growth factor (R&D system, Minneapolis, MN, USA), and 100 units/mL penicillin plus 100 ug/mL streptomycin (Life Technologies, NY, USA). For differentiation, GSCs were cultured in NS-A medium supplied with 10% fetal bovine serum. Anti-BIRC3 antibody was obtained from R&D system; anti-b-actin IgG-HRP was obtained from Santa Cruz Biotech; and anti-SMAD1, anti-SMAD5, anti-p-SMAD1/5, Goat anti-Rabbit IgG-HRP and Goat anti-mouse IgG-HRP were obtained from CellSignal.

### 4.2. Gene Overexpression in GBM Cell

Human BIRC3 expression and empty vector constructs were obtained from Genecopoeia (Rockville, MD, USA). A single bacteria clone was picked from a freshly streaked LB plate containing100 μg/mL ampicillin and inoculated to a culture of 5 mL LB medium containing 100 μg/mL ampicillin, which was then incubated for 16 h at 37 °C with vigorous shaking. Plasmid was purified using QuickLyse Miniprep Kit (Qiagen, Germantown, MD, USA). U251, U87 cells and GSCs (2 × 10^5^) was seeded in 6-well plate 24 h before transfection. BIRC3 expression plasmid was transfected by Lipofectmine 3000 kit (for U251 and U87, ThermoFisher, Waltham, MA, USA) and Lipofectmine Stem reagent (for GSCs, ThermoFisher, Waltham, MA, USA) following manufacture’s protocol. The cells were then incubated at 37 °C in a CO_2_ incubator for 48 h. G418 sulfate (500 μg/mL) was used for selection 48 h after transfection and the G418 sulfate concentration was then reduced to 200 μg/mL 7 days later for maintenance. The overexpression of BIRC3 was verified by western blot.

### 4.3. Gene Silencing by CRISPR/Cas9 System

CRISPR/Cas9 vectors lentiCRISPR-v2-puro was obtained from Addgene. sgRNA targeting human/mouse BIRC3 and sgRNA control were cloned into lentiCRISPR-v2-puro. Human BIRC3 targeting forward primer: CACCGTATTTCAGTTCAAACGTGT, reverse primer: AAACACACGTTTGAACTGAAATAC; Mouse BIRC3 targeting forward primer: CACCGTTCCGGCGCGCCGAGTCCTT, reverse primer: AAACAAGGACTCGGCGCGCCGGAAC; control sgRNA cloning forward primer: CACCGCACTCACATCGCTACATCA, reverse primer: AAACTGATGTAGCGATGTGAGTGC. Lentivirus was packed by 293T cells through 2^nd^ generation lentivirus packaging system. U251, U87, CT-2A cells and GSCs were next infected with Lenti-sgBIRC3-puro or Lenti-sgControl-puro followed by extensive selection with 1μg/mL puromycin (InvivoGen, San Diego, CA, USA). To confirm CRISPR silencing efficiency, we harvested protein from cell lysis and tested them with western blot.

### 4.4. Real-Time PCR

Total RNA was extracted using RNeasy mini-prep kit (Qiagen, Germantown, MD, USA). RNA was quantified with Nanodrop 2000 (Thermo Scientific, Waltham, MA, USA). cDNA was synthesized using 1ug total RNA with the iScript cDNA Synthesis kit (Bio-Rad, Hercules, CA, USA). Real-time PCR was performed using iQ SYBR green Supermix buffer system (Bio-Rad, Hercules, CA, USA) and the Bio-Rad CFX96 Touch Real-Time PCR Detection system. Human CD133 forward primer: ACTCCCATAAAGCTGGACCC, reverse primer: TCAATTTTGGATTCATATGCCTT; human ABCG2 forward primer: AGCAGCAGGTCAGAGTGTGG, reverse primer: GATCGATGCCCTGCTTTACC; human ALDH1A3 forward primer: TGGATCAACTGCTACAACGC, reverse primer: CACTTCTGTGTATTCGGCCA; human BMP4 forward primer: GCCGGAGGGCCAAGCGTAGCCCTAAG, reverse primer: CTGCCTGATCTCAGCGGCACCCACATC; human GAPDH was used as the internal control, GAPDH forward primer: ACCACAGTCCATGCCATCAC, reverse primer: TCCACCACCCTGTTGCTGT. Mouse CD133 forward primer: TTGGTGCAAATGTGGAAAAG, reverse primer: ATTGCCATTGTTCCTTGAGC; mouse ABCG2 forward primer: CAGTTCTCAGCAGCTCTTCGAC, reverse primer: TCCTCCAGAGATGCCACGGATA; mouse BMP4 forward primer: GCCGAGCCAACACTGTGAGGA, reverse primer: GATGCTGCTGAGGTTGAAGAGG; mouse GAPDH was used as the internal control, GAPDH forward primer: ATGGTGAAGGTCGGTGTGA, reverse primer: AATCTCCACTTTGCCACTGC. The PCR program was as follow: 95 °C 10 min, 1 cycle; 95 °C 15 s, → 60 °C 30 s → 72 °C 30 s, 40 cycles; 72 °C 10 min, 1 cycle.

### 4.5. Western Blot Analysis

50–100 μg of heat-denatured proteins were loaded on 4–15% precast polyacrylamide gel (Bio-Rad, Hercules, CA, USA). The proteins were then transferred to PVDF membranes (Bio-Rad, Hercules, CA, USA), which were blocked with 5% non-fat milk solutions for 1 hour at room temperature. The target proteins were then detected by the primary antibody at 4 °C overnight, washed with 0.1% Tween-TBS and incubated with appropriate secondary antibody for 1 hour at room temperature. The membranes were then washed, and the target proteins were detected with luminol reagent and X-ray film (Santa Cruz Biotechnology, Dallas, TX, USA).

### 4.6. Tumor Sphere Formation Assay

U251 and U87 Cells were collected, counted, and seeded in DMEM/F12 medium with B27 supplement, 20 ng/mL human recombinant epidermal growth factor (EGF, ThermoFisher, Waltham, MA, USA), and 10 ng/mL human recombinant basic fibroblast growth factor (FGF-2, Sigma-Aldrich, St. Louis, MO, USA), 100 units/mL-penicillin-100 ug/mL streptomycin (Life Technologies, NY, USA). Differentiated GSCs were collected, counted, and seeded in NS-A medium, 20 ng/mL human recombinant epidermal growth factor (EGF, ThermoFisher, Waltham, MA, USA), and 10 ng/mL human recombinant basic fibroblast growth factor (FGF-2, Sigma-Aldrich, St. Louis, MO, USA), 100 units/mL-penicillin-100 ug/mL streptomycin (Life Technologies, NY, USA). The cells were subsequently cultured in ultra-low attachment 6-well plates (Corning, NY, USA) at a density of 2000–5000 cells/well. Half of the culture medium was replaced or supplemented with additional growth factors twice a week. To propagate spheres in vitro, the cells were collected by gentle centrifugation, dissociated by Accutase (StemCell Technologies, Vancouver, Canada) into single-cell suspensions and cultured to allow the regeneration of spheres. Third-generation spheres were used for all subsequent experiments except siRNA knockdown sphere formation. For siRNA knockdown cells, first-generation spheres were used. The total number of tumor spheres was counted following 10 days of culture. Images are taken under 4× magnification or 10× magnification observation.

### 4.7. Immunocytochemistry Analysis

Cells were seeded onto poly-L-lysine–coated slides and cultured for 24 h. Then, cells were fixed using Cytofix/Cytoperm (BD Biosciences, San Jose, CA, USA) according to the manufacturer’s protocol. Cells were washed in phosphate-buffered saline (PBS), blocked with PBS buffer containing 2% BSA and 0.1% TritonX-100 for 1 h, and incubated with Alexa Fluor 488 anti-Nestin (BioLegend, San Diego, CA, USA) overnight. Next, cells were washed with PBS and slides were mounted onto coverslips over a drop of Vectashield mounting medium with DAPI (Vector Laboratories, Burlingame, CA, USA). Cells were examined with an automated Zeiss Observer Z.1 inverted microscope through a 63X/1.4NA objective and DAPI and FITC filters. Multi-channel images were captured using the AxioCam MRm3 CCD camera and Axiovision version 4.7 software suite (Carl Zeiss Inc, Oberkochen, Germany).

### 4.8. siRNA Knockdown

U251 and U87 cells were transfected with predesigned BMP4 small interfering RNA (siRNA; 30 nM, Millipore-Sigma, St. Louis, MO, USA) or control siRNA (30 nM, Millipore-Sigma, St. Louis, MO, USA) using Lipofectamine RNAiMAX Transfection Reagent (ThermoFisher, Waltham, MA, USA). Briefly, one day prior to transfection, the cells were seeded in 6-well plate (2 × 10^5^) with 10% FBS DMEM without antibiotics. siRNAs were prepared according to the manufacturer’s instructions and added to the cells. For tumor sphere formation the cells were then exposed to tumor sphere formation culture medium after 24 h.

### 4.9. Mice and GBM Orthotopic Xenograft Model

Female NCRNU athymic mice of 6–8 weeks were ordered from Taconic Biosciences. All animals were housed in the American Association for Laboratory Animal Care-accredited Animal Resource Center at Moffitt Cancer Center. All animal procedures and Experiments were carried out under protocols approved by the Institutional Animal Care and Use Committee of the University of South Florida and Moffitt Cancer Center. All animal studies were performed in accordance with relevant guidelines and regulations of University of South Florida and Moffitt Cancer Center. Tumors were established by injecting 2 × 10^5^ U251 control, BIRC3 overexpression and BIRC3 knockout cells in a 4 μL volume of PBS in the right striatum of mice (n = 5/group) on a Stoelting Digital Stereotaxic Instrument (Stoelting, IL, USA). The tumor progression was monitored by MRI (Bruker Biospec 7T, Billerica, MA, USA) every week. For survival studies, animals were followed until they lost 20% of body weight or had trouble ambulating, feeding, or grooming.

### 4.10. Immunohistochemistry

Tumor samples were fixed with 10% neutral-formalin buffer for 72 h. The samples were then dehydrated, paraffin-embedded and sectioned. Sections were dewaxed, treated with 3% H_2_O_2_ for 10 min and incubated with anti-BIRC3 antibody (1:100 dilutions) overnight at 4 °C. Biotinylated secondary antibody (1:200 dilutions) was added at room temperature for 1 h, followed by the incubation with ABC-peroxidase for additional 1 h. After washing with Tris-buffer, the sections were incubated with DAB (3, 30 diaminobenzidine, 30 mg dissolved in 100 mL Tris-buffer containing 0.03% H_2_O_2_) for 5 min, rinsed in water and counterstained with hematoxylin.

### 4.11. Bioinformatics and Statistics

The IVY data was downloaded and log2 transformed. The TCGA GBM samples was extracted from the normalized and debatched PanCan RNA-Seq data and log2 transformed. Student’s t-test (for 2 condition experiments) and ANOVA (for multiple condition experiments) was employed. Survival was assessed using Kaplan-Meier analysis with statistical comparisons made by log rank (Mantel-Cox) test. All statistical tests were considered significant at *p* < 0.05. * means *p* < 0.05.

## Figures and Tables

**Figure 1 ijms-23-00297-f001:**
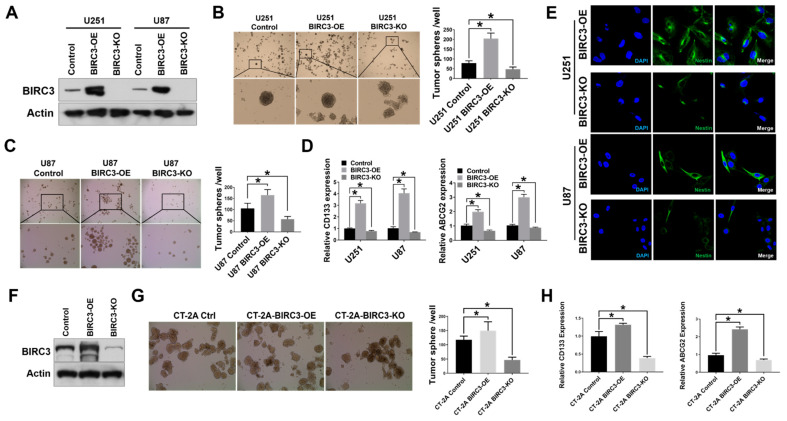
Expression of BIRC3 correlates with stem cell markers expression and self-renewal in both human and mouse GBM cells (**A**). Protein expression of BIRC3 in U251 and U87 GBM cells. Each cell line includes control, BIRC3 overexpression (BIRC3-OE) and BIRC3 knockout (BIRC3-KO) groups. Specific antibodies as indicated. β-actin acts as internal control. (**B**,**C**). Control, BIRC3-OE or BIRC3-KO of U251 and U87 cells were seeded in 6 well plates and cultured in neurosphere formation medium. The number of neurospheres were observed and calculated under microscope. (**B**): Representative images are under 4× magnification (top raw) and 20× magnification (bottom raw). (**C**): Representative images are under 4× magnification (top raw) and 10× magnification (bottom raw). n = 5, * *p* < 0.05. (**D**). CD133 and ABCG2 mRNA expression analyzed by real-time PCR in U251/U87 control, BIRC3-OE and BIRC3-KO cells. n = 3, * *p* < 0.05. (**E**). Immunofluorescence staining of Nestin in U251/U87 BIRC3-OE and BIRC3-KO cells. Blue: DAPI; Green: Nestin. (**F**). Protein expression of mBIRC3 in CT-2A mouse GBM cells including control, BIRC3-OE and BIRC3-KO groups. Specific antibodies as indicated. β-actin acts as internal control. (**G**). Control, BIRC3-OE or BIRC3-KO of CT-2A cells were seeded in 6 well plates and cultured in neurosphere formation medium. The number of neurospheres were observed and calculated under microscope. Representative images are under 10× magnification. *n* = 5, * *p* < 0.05. (**H**). Mouse CD133 and ABCG2 mRNA expression were analyzed by real-time PCR in CT-2A cells. *n* = 3, * *p* < 0.05.

**Figure 2 ijms-23-00297-f002:**
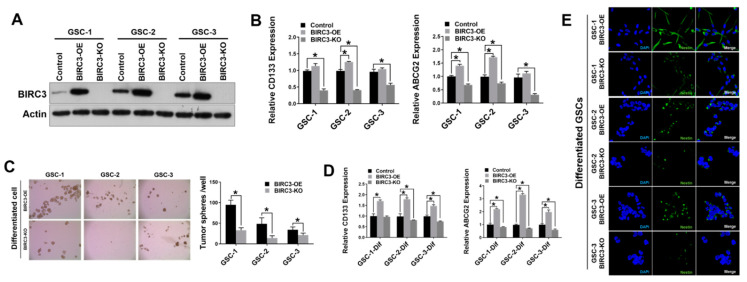
Human GBM stem cell self-renewal is regulated by BIRC3 expression. (**A**). Protein expression of BIRC3 in three different GSCs. Each GSC includes control, BIRC3-OE and BIRC3-KO groups. Specific antibodies as indicated. β-actin acts as internal control. (**B**). CD133 and ABCG2 mRNA expression analyzed by real time PCR in control, BIRC3-OE and BIRC3-KO GSCs. *n* = 3, * *p* < 0.05. (**C**). Control, BIRC3-OE or BIRC3-KO of differentiated GSCs were seeded in 6-well plate and cultured in neurosphere formation medium. The number of neurospheres were observed and calculated by microscope. Representative images are under 4× magnification. *n* = 5, * *p* < 0.05. (**D**). CD133 and ABCG2 mRNA expression analyzed by real time PCR in control, BIRC3-OE and BIRC3-KO of differentiated GSCs. *n* = 3, * *p* < 0.05. (**E**). Immunofluorescence staining of Nestin in differentiated BIRC3-OE and BIRC3-KO GSCs.

**Figure 3 ijms-23-00297-f003:**
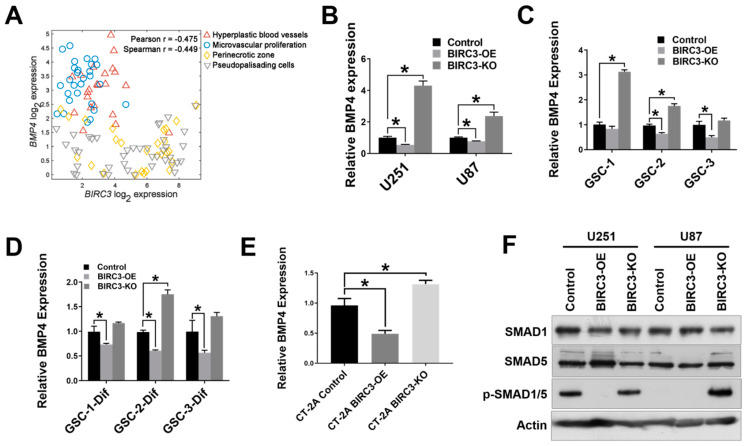
BIRC3 directs BMP4 signaling inhibition in GBM. (**A**). Analysis of correlation between *BIRC3* and *BMP4* expression in different regions of GBM using IVY dataset. Analyzed regions include hyperplastic blood vessels, microvascular proliferation region, perinecrotic zone and pseudopalisading cells region. (**B**–**D**). Human BMP4 mRNA expression analyzed by real time PCR in control, BIRC3-OE and BIRC3-KO cells including U251/U87 GBM cell lines, GSCs and differentiated GSCs. *n* = 3, * *p* < 0.05. (**E**). Mouse BMP4 mRNA expression analyzed by real time PCR in control, BIRC3-OE and BIRC3-CT-2A cells. *n* = 3, * *p* < 0.05. (**F**). Protein expression of SMAD1, SMAD5 and phosphorylated SMAD1/5 in U251/U87 control, BIRC3-OE and BIRC3-KO cells. Specific antibodies as indicated. β-actin acts as internal control.

**Figure 4 ijms-23-00297-f004:**
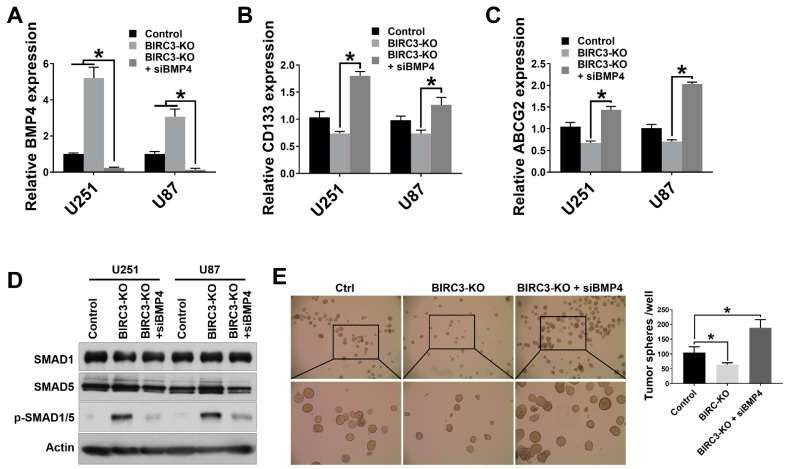
BIRC3 impacts GBM cell self-renewal and is dependent on BMP4 suppression. (**A**). Human BMP4 mRNA expression analyzed by real time PCR in control, BIRC3-KO, and BMP4-siRNA silenced BIRC3-KO U251/U87 GBM cell lines. *n* = 3, * *p* < 0.05. (**B**). Human CD133 mRNA expression analyzed by real time PCR in control, BIRC3-KO, and BMP4-siRNA silenced BIRC3-KO U251/U87 GBM cell lines. *n* = 3, * *p* < 0.05. (**C**). Human ABCG2 mRNA expression analyzed by real time PCR in control, BIRC3-KO, and BMP4-siRNA silenced BIRC3-KO U251/U87 GBM cell lines. *n* = 3, * *p* < 0.05. (**D**). Protein expression of SMAD1, SMAD5 and phosphorylated SMAD1/5 in control, BIRC3-KO, and BMP4-siRNA silenced BIRC3-KO U251/U87 GBM cell lines. Specific antibodies as indicated. β-actin acts as internal control. (**E**). Control or BIRC3-KO of U251 cells were seeded in 6 well plates and cultured in neurosphere formation medium. The BIRC3-KO cells had been treated with control siRNA and BMP4 siRNA separately 1 day before seeding. The number of neurospheres were observed and calculated by microscope. Representative images are under 4× magnification (top raw) and 10× magnification (bottom raw). *n* = 5, * *p* < 0.05.

**Figure 5 ijms-23-00297-f005:**
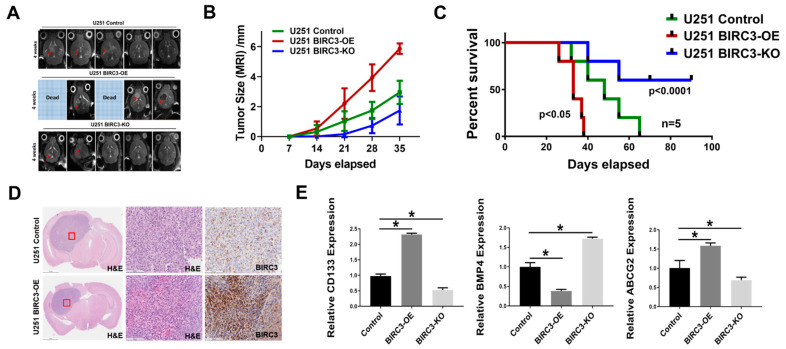
BIRC3 influences tumor initiation and progression in GBM orthotopic xenograft model. GBM Intracranial models with control, BIRC3-OE and BIRC3-KO U251 cells. (**A**). Horizontal axial MRI scan of mouse brain tumors 4 weeks after implantation. Two of BIRC3-OE mice were already dead at 4 weeks. (**B**). Tumor size calculation from MRI scan. *n* = 5. (**C**). Kaplan-Meier survival curve of U251 control BIRC3-OE and BIRC3-KO intracranial injection mice. n = 5 mice/group. (**D**). Mice were sacrificed at different timepoints and brain tissues of U251 control and BIRC3-OE groups were fixed in 10% neutral formalin. H&E staining and BIRC3 immunohistochemistry was performed as described in the Material and Methods Section 4. Five mice were included in this histological study and similar results were observed in each animal. (**E**). When mice were sacrificed, part of tumor tissues were isolated. mRNA from tumor tissues were extracted. BMP4, CD133 and ABCG2 mRNA expression analyzed by real-time PCR in extracted tumor tissues. *n* = 3, * *p* < 0.05.

## Data Availability

The data that support the findings of this study are available from the corresponding authors upon reasonable request.

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
