# Peer review of "A Novel Role of BIRC3 in Stemness Reprogramming of Glioblastoma"

_ijms, 2021, doi:10.3390/ijms23010297_

Round 1

Reviewer 1 Report

The manuscript titled “A novel role of BIRC3 in stemness reprogramming of glioblastoma” by Qiong Wu et al. demonstrated that BIRC3 drives stemness reprogramming of human GBM cell lines, mouse GBM cell line and patient derived GBM stem cells (GSCs) through regulation of BMP4 signaling axis. The manuscript is well-balanced with good writing.

Minor Point:

  1. The sphere formation ability of U251 cells may be weak. Seeding more cells may show the difference of sphere formation after BIRC3 knockout compared with control cells.
  2. It seems that BIRC3 may affect the mRNA level of BMP4. Please try to analyze the expression correlation between BIRC3 and BMP4, and BMP4 signaling with already established tissue RNA sequencing data.
  3. The potential of targeting BMP4 to treat glioma has been summarized. It will be better to mention it and cite the following paper in this manuscript.

Nayak, S.; Mahenthiran, A.; Yang, Y.; McClendon, M.; Mania-Farnell, B.; James, C.D.; Kessler, J.A.; Tomita, T.; Cheng, S.-Y.; Stupp, S.I.; Xi, G. Bone Morphogenetic Protein 4 Targeting Glioma Stem-Like Cells for Malignant Glioma Treatment: Latest Advances and Implications for Clinical Application. Cancers 202012, 516. https://doi.org/10.3390/cancers12020516

Author Response

Point-point response to the reviewers

Reviewer 1

The manuscript titled “A novel role of BIRC3 in stemness reprogramming of glioblastoma” by Qiong Wu et al. demonstrated that BIRC3 drives stemness reprogramming of human GBM cell lines, mouse GBM cell line and patient derived GBM stem cells (GSCs) through regulation of BMP4 signaling axis. The manuscript is well-balanced with good writing.

We thank the reviewer for the positive feedback and we have addressed the questions and concerns raised during the review point by point below.

Minor Point:

  1. Q: The sphere formation ability of U251 cells may be weak. Seeding more cells may show the difference of sphere formation after BIRC3 knockout compared with control cells.

R: We thank the reviewer for the great suggestion. We have seeded 5000 cells per well for the new sphere formation assay with U251 cells. As shown in the revised Fig. 1B, we observed more sphere formation. The difference between U251 BIRC3-OE and wild type control cells becomes more significant (>3 fold). BIRC3 knockout led to a significant decrease in neurosphere formation compared to control, but not to same extent that BIRC3-OE impacted neurosphere formation.

  1. Q: It seems that BIRC3 may affect the mRNA level of BMP4. Please try to analyze the expression correlation between BIRC3 and BMP4, and BMP4 signaling with already established tissue RNA sequencing data.

R: Thank you very much for this valuable suggestion. We agree that analyzing the correlation between BIRC3 and BMP4 in patient RNA-Seq data could provide clarity on clinical relevance. We have first analyzed BIRC3 and BMP4 expression within TCGA GBM dataset which looks at bulk tumor and does not take in account regional intra-tumoral heterogeneity in GBM. As shown in Fig. S3, BIRC3 and BMP4 showed low correlation in whole patient tumor tissues TCGA data. In order to account for regional intra-tumoral heterogeneity, we evaluated RNA-Seq data from the IVY GBM dataset which contains RNA-Seq data from several regions of GBM - including vascular niches (hyperplastic blood vessels region and microvascular proliferation region) and hypoxic niches (perinecrotic zone and pseudopalisading region). The hypoxic niche has been recognized as a region for stemness reprogramming.  In all regions, BIRC3 and BMP4 demonstrated a robust negative correlation (Fig. 3A).  The hypoxic niche demonstrated high BIRC3/low BMP4 (Fig. 3A). Those results indicate the impact of GBM heterogeneity on BIRC3-BMP4 regulation and will undoubtedly impact on GBM stem cell maintenance.

  1. Q: The potential of targeting BMP4 to treat glioma has been summarized. It will be better to mention it and cite the following paper in this manuscript.

R: Thank you very much for this suggestion. We strongly agree that since we target BIRC3-BMP4 signaling here, we should cite such publication in our manuscript. Now we have cited this paper below in our revised manuscript (Ref 32). [Nayak, S.; Mahenthiran, A.; Yang, Y.; McClendon, M.; Mania-Farnell, B.; James, C.D.; Kessler, J.A.; Tomita, T.; Cheng, S.-Y.; Stupp, S.I.; Xi, G. Bone Morphogenetic Protein 4 Targeting Glioma Stem-Like Cells for Malignant Glioma Treatment: Latest Advances and Implications for Clinical Application. Cancers 2020, 12, 516. https://doi.org/10.3390/cancers12020516]

Reviewer 2 Report

1. Why authors used stem cells markers (CD133 and ABCG2) but others glioblastoma stem cell markers (for example, ALDH1A3, CD15, CD27L and others)? 2. In "Methods" authors did not indicated which Probe or Fluorescent dye used in Real-time PCR (when estimated expression of markers CD133, ABCG2, BMP4)? 3. Why in this case authors used real-time PCR for detection of stem cells markers but not flow cytofluorimetry? 4. What is the further application of these findings? In glioblastoma therapy only?

Author Response

Point-point response to the reviewers

Reviewer 2

Comments and Suggestions for Authors

  1. Q: Why authors used stem cells markers (CD133 and ABCG2) but others glioblastoma stem cell markers (for example, ALDH1A3, CD15, CD27L and others)?

R: We thank the reviewer of this comment and suggestion. According to our previous published data (Ref 16), BIRC3 is strongly associated with mesenchymal phenotype in GBM. High BIRC3 expressing cells show higher expression of mesenchymal GBM markers, including mesenchymal-stemness marker. Data from several studies (Ref 21) reveal that both CD133 and ABCG2 are linked to mesenchymal-like GBM stem cells and further those two markers showed very robust expression in BIRC3 expressing cells in our study. Besides those two gene markers, ALDH1A3 is also a mesenchymal-like stemness marker in GBM (Ref 21). Following the reviewer’s suggestion, we evaluated ALDH1A3 expression in both GBM cell lines (U251 and U87) and GSCs, and ALDH1A3 revealed a very similar expression pattern as CD133 and ABCG2. We have placed those results as supplemental data in the revised manuscript, and also included primers sequence in “Methods” section. The GBM stem cell markers demonstrate that BIRC3 positively impact GBM cell stemness. It has been reported that GSCs significantly express CD15 at the cell surface along with CD133. CD133 expression could also reflect CD15 status. For CD27L (CD70), its role and expression in GBM is complicated, and its expression varies a lot in different GSCs and hard to detect in some of our GSCs.  

  1. Q: In "Methods" authors did not indicated which Probe or Fluorescent dye used in Real-time PCR (when estimated expression of markers CD133, ABCG2, BMP4)?

R: We thank the reviewer for this important reminder. We used SYBR green buffer system in Real-time PCR, and we added this in the “Methods” section.

  1. Q: Why in this case authors used real-time PCR for detection of stem cells markers but not flow cytofluorimetry?

R: We thank the reviewer for the comment. Real-time PCR demonstrates those stem cells makers expression level, which could estimate the cells real status and the potential to become “stem-like” cells. Flow cytometry will only identify stem marker, for example CD133, as positive or negative cells, but ignore the transition status between them. The mRNA expression will directly reflect the impact of BIRC3 on those downstream stemness-related signaling and gene expression. The stem gene marker expression will further precisely reflect the cell responses and change(s) to BIRC3 expression level alteration, but not only the end point phenotype (positive or negative cells).

  1. Q: What is the further application of these findings? In glioblastoma therapy only

R: We appreciate this comment from reviewer. GBM remains a very challenging tumor to treat secondary to intra-tumoral heterogeneity and stemness reprogramming.  Therefore, it is of utmost importance to identify and target mechanisms of stemness reprogramming in GBM. Here we have identified a novel noncanonical function of BIRC3 as a regulator of GBM stemness reprogramming through suppression of BMP4 signaling axis. Our future studies will examine the mechanistic underpinnings of BIRC3-mediated stemness to identify pathway components amenable to pharmacologic targeting. Besides, many researchers are developing BIRC3 specific inhibitors including small-molecule SMAC mimetics which could have potential applications in targeting stemness reprogramming in GBM.